# A Privacy-Preserving Approach to Effectively Utilize Distributed Data for Malaria Image Detection

**DOI:** 10.3390/bioengineering11040340

**Published:** 2024-03-30

**Authors:** Amer Kareem, Haiming Liu, Vladan Velisavljevic

**Affiliations:** 1School of Computer Science and Technology, University of Bedfordshire, Luton LU1 3JU, UK; vladan.velisavljevic@beds.ac.uk; 2School of Electronics and Computer Science, University of Southampton, Southampton SO17 1BJ, UK; h.liu@soton.ac.uk

**Keywords:** malaria images, machine learning, federated learning, privacy preserving, medical image detection

## Abstract

Malaria is one of the life-threatening diseases caused by the parasite known as Plasmodium falciparum, affecting the human red blood cells. Therefore, it is an important to have an effective computer-aided system in place for early detection and treatment. The visual heterogeneity of the malaria dataset is highly complex and dynamic, therefore higher number of images are needed to train the machine learning (ML) models effectively. However, hospitals as well as medical institutions do not share the medical image data for collaboration due to general data protection regulations (GDPR) and the data protection act (DPA). To overcome this collaborative challenge, our research utilised real-time medical image data in the framework of federated learning (FL). We have used state-of-the-art ML models that include the ResNet-50 and DenseNet in a federated learning framework. We have experimented both models in different settings on a malaria dataset constituting 27,560 publicly available images and our preliminary results showed that the DenseNet model performed better in accuracy (75%) in contrast to ResNet-50 (72%) while considering eight clients, while the trend was observed as common in four clients with the similar accuracy of 94%, and six clients showed that the DenseNet model performed quite well with the accuracy of 92%, while ResNet-50 achieved only 72%. The federated learning framework enhances the accuracy due to its decentralised nature, continuous learning, and effective communication among clients, as well as the efficient local adaptation. The use of federated learning architecture among the distinct clients for ensuring the data privacy and following GDPR is the contribution of this research work.

## 1. Introduction

During study of related work, we have observed that, over recent years, advancements in the field of artificial intelligence (AI) have brought a great revolution in the field of medical sciences. It has been demonstrated as an effective way of deployment to detect diseases through CXR, city scan, ultrasound, and other mediums. Researchers are using artificial intelligence for diagnosis and detection. Improving computer vision in AI increases the research interest in medical diagnosis. Regarding medical image detection, AI techniques such as convolution neural networks (CNN) have been used to classify CXR, whether the disease is present or not. While considering AI in the medical field, significant amounts of research have been done that include abnormal pattern recognition [1,2], biometric detection [3], trauma valuation [4], and diabetes detection [5]. The importance of medical image detection has brought us to work in devising effective methods for detecting diseases including pneumonia, malaria, and brain tumors. Malaria is one of the fatal diseases known to be transmitted by mosquitoes. According to the World Malaria report, 241 million malaria cases were reported in 2020, which is higher than 2019 when it was recorded as 227 million. Over 600,000 individuals died due to malaria in 2020, where an 80% fatality trend was observed among children below 5 years of age [6]. Therefore, it is important to understand the severity of this disease, and devise solutions to tackle the disease.

### 1.1. Research Background

Medical image classification is quite a complex task in nature. The medical images gathered from different sources that include chest X-ray, CT scan, or microscopic images constitute higher dimension and are complex in nature. Medical images are higher in spatial resolution as well as complex in patterns [7]. It is also quite challenging for researchers to utilise the real-time image data as it constitutes patients’ bio-details. Moreover, the class imbalance between the normal images and infected images is also one of the major challenges in medical image classification. Different state-of-the-art techniques have been used to enhance medical image detection that include deep learning, supervised learning, unsupervised learning and reinforcement learning. However, the lack of data availability has always been challenging to perform the machine learning modelling effectively. Therefore, considering the complexity, as well as the data imbalance of the medical images, it is important to access the real-time data to meet the requirements of data availability. However, due to general data protection and regulation (GDPR), data cannot be shared to the third party [8]. Our motivation and the scope of this research for malaria image detection is inspired by employing the ML techniques in the privacy-preserving framework of federated learning for utilising real-time data, while following GDPR rules and regulations.

### 1.2. Problem Statement and Rationale of the Research

Considering the background of our research, the following are the problem statements:Medical images are highly heterogeneous compared to the normal images, therefore it is quite challenging to perform ML modelling on the limited data available [9]. To compensate this issue, we are required to form a collaborative framework that allows multiple hospitals and medical institutions to share data in a privacy-preserving manner.Lab-based data or synthesized data are limited to perform effective ML modelling [10], therefore we were required to have vast quantity of data that can be fulfilled by using the live stream of real-time data.It can be possible to use the real-time data to fulfil the data availability problem, however due to general data regulation and protection (GDPR), data sharing cannot be possible. Therefore, our research is inspired to use the privacy-preserving framework of federated learning (FL) to allow the data sharing while following GDPR.

Based on the above problem statements, the following research questions (RQs) can be concluded:Is it possible to utilise real-time malaria image data by collaboration of different hospitals and medical institutes in a privacy-preserving manner?What are state-of-the-art machine learning approaches for effective malaria image detection?

The above research questions are further elaborated in the following section.

### 1.3. Significance of the Research

Considering the potential of the proposed research work, the medical imaging industry has a huge need to alter the procedures for disease detection. The following are some of the important points with regard to the significance of our research.

Improved security and compliance: The proposed research work provides huge potential considering the privacy and security of medical imaging. The research follows the guidelines as per GDPR and DPA for data security that will be a great revolution in the medical industry.Enhanced diagnostic capacity: The research framework that constitutes the hybrid model will ensure the security of data and the accuracy efficiency of disease detection that will ultimately result in early diagnosis and treatment.Facilitating collaboration: Research will promote an innovative culture that allows the mutual collaboration of hospitals and medical institutes to achieve the improved advancements.Benchmark for potential innovation: Based on the study analysis, it will guide the future scope of innovation in medical imaging for researchers. The proposed research can be a benchmark for the future development of this idea that can be mutually beneficial for medical institutes.Scalable and flexible framework: The research illustrates the use of a CNN-based pre-trained model on the federated learning framework that is highly scalable towards the multiple types of medical images, and provides a robust and enhanced solution for medical image detection.Economic influence: Research will bring important changes to ease the economic impact, such as early detection. It will ultimately bring about early disease detection, saving costs and resources in the medical industry.Global extent and convenience: Using the FL framework as in the proposed research will ensure data privacy, allowing data from diverse sources to enhance machine learning models’ learning capability.

It can be seen from the significance of the proposed research work that the industrial implications are massive. It reflects the current situation in the medical field and the future scope that will allow medical institutes to collaborate in a single platform for improved diagnoses and early treatment. The research will also bring academic institutes together to develop an innovative solution to medical image detection by collaborating with the image data of the medical center in a privacy-preserving manner.

### 1.4. Contribution to Knowledge

Our research contribution involves the use of a hybrid approach of the federated learning framework and CNN-based pre-trained models of DenseNet and ResNet-50 for malaria image detection. It involves the mutual collaboration of hospitals and medical institutes, while sharing data in a privacy-preserving manner. The novelty of this research work reflects the approach of data sharing while following GDPR rules.

### 1.5. Paper Organisation

This article constitutes an introduction part, followed up with the literature review where we have critically reviewed various state-of-the-art literature. After the literature review, follows the methodology that includes research model design, and configuration of models in federated learning architecture. It further includes data gathering and exploratory data analysis (EDA). After the research model design, it includes malaria experiment and results in four, six and eight client settings. It further involves the section of significance test, followed up with the conclusion and references.

## 2. Literature Review

This section demonstrates the different state-of-the-art techniques that are used for medical image detection. We have critically analysed the previous work and highlighted the limitations as well as reasons for the selection of CNN-based models.

Researchers have used the CNN pre-trained model of ResNet-50 for the detection of the diabetic retinopathy, which is the major cause of blindness in diabetic patients [11]. The researchers followed the optimal steps for image pre-processing and augmentation. In the experiment, dataset of 3762 images was used, among them 1855 were healthy ones and 1907 were infected from the Eyepacs. The authors have compared the work with the other state-of-the-art literature and found that the performance of the ResNet-50 is effective when the image pre-processing is enhanced, as the model performance can vary depending on the input images. In the machine learning context, with regards to the image detection, the author has demonstrated the outcome of the experiments while achieving the accuracy of 0.9802 in the binary classification. It gives a clear view of using the dataset giving better augmentation and data cleaning, to let the model perform effectively.

Another experiment was performed while comparing the model of ResNet-50 and VGG16 for the detection of the COVID-19 [12]. In the experiment, the dataset of “COVID-19 Radiography Database” was used that was obtained from the Kaggle. The dataset constitutes of the images over 10 k, among them 3600 were infected. After performing the data cleaning and data pre-processing, the authors have individually used the ResNet-50 and VGG16 model. The results of the experiments have demonstrated that the performance of ResNet-50 stands out in contrast to the VGG16. The achieved accuracy on the experiment for ResNet-5050 was 88% while, on the other hand, the achieved accuracy for the VGG16 was 85% for the detection of the COVID-19. In the same experiment, the precision achieved was 100% for ResNet-50, and 84% for the VGG network. The authors have suggested that the overall accuracy of the model performance can be enhanced while using the hyperparameter tunning.

Research was conducted while using the ResNet-50 model on the COVID-19 dataset [13]. The CT scan images were used for the experiment. The dataset constituting over 5 K CT scan images was utilised. The researchers realised that the quite larger number of datasets was similar to the pre-training dataset, therefore the CNN-based models can be less effective. To cope with this challenge, researchers were urged to perform fine tuning of the model on the training dataset to enhance the performance of the model and also reduce the time consumption of the model training. While using the same optimiser, the authors achieved the accuracy of 88% on the normal CNN model with the default tuning, however, when the authors used the similar optimiser on the tuned CNN model, the accuracy went up by 6%, which is 94%. The experiment performed by the authors clearly distinguishes the use of different hyperparameter tuning to effectively increase the model accuracy for the disease prediction and also improve the time consumption. As per our proposed model for using the real-time data, where the amount of data and time required to train the model will be challenging, it is important to define the model into a state of fine tuning to obtain the best possible outcome.

The researchers have proposed the use of an enhanced method for tuberculosis (TB) detection from the CXR while using the DenseNet model [14]. The experiment was performed while using the wider framework of the DenseNet (WDnet), which was based on the Convolutional Block Attention Module (CBAM). In the experiment, a dataset was used from multiple repositories and then combined together. The combination of the repositories has produced images up to 5 k. Among the image dataset, 1094 were classed as infected images while the remaining were classed as normal. While comparing the model with other literature, the researchers demonstrated the evaluation of the experiments by producing the accuracy of 98.80%. This research has highlighted the use of different epochs to understand the best possible outcome of the experiment for medical image detection.

The study was conducted to highlight the importance of using a deep learning approach for effective medical image detection [15]. In the literature, authors have analysed the different machine learning models for medical image classification and used the benchmark CT scan dataset to compare the results. The authors have raised the concern of challenges in using the deep learning model for training the dataset. Also, it has been mentioned in the research that the new dataset for the pre-trained model could be complex, therefore it is important we understand what is lacking, and use the effective approach to obtain the least false positive or negative results. This research gives us an understanding of using the deep learning model for larger amount of datasets to understand the maximum number of patterns in the data, especially in the medical images. Our proposed solution of using the real-time data in the federated learning framework gives a clear way forward to utilise the constant stream of data from different hospitals and medical institutions, that would eventually help to make deep learning models more capable of recognising different diseases and effective classification accuracy.

The experiment was conducted to analyse the performance of the neural network in the detection of SARS-CoV-2 virus [16]. The research was performed while using the CNN model of VGG-16, VGG-19, ResNet 50, Inception v3, DenseNet, XceptionNet, and MobileNet v2. The dataset used in the experiment consists of 1252 COVID and 1229 Non-COVID CT scan images. While comparing the different CNN models, the authors defined the use of proposed CNN model that has produced the accuracy of 92%. The main idea of the research was to identify the different model performance, do the alteration on the CNN pre-trained model and produce the customised CNN approach for effective classification. The results of the experiment have been demonstrated in the paper that gives the clear picture of the use of customised CNN model. Although the customised CNN approach has produced the classification of the SARS-CoV-2 virus, the proposed model of the authors could be challenging when followed up with the higher number of images. The current experiment demonstrates the limited number of images. While considering our proposed method of using the real-time data in the federated learning framework, it possesses limitations, the idea is to use the model that is pre-trained and scalable. The authors’ given experiment is effective in the limited number of datasets, however higher numbers can adversely impact the performance of the model.

In another experiment [17], the researchers performed experiments for the detection of the pneumonia disease while using the ResNet. In the experiment, authors have used the different version of the ResNet to compare the results. The researchers have used the optimised attention mechanism to enhance the performance of the model that includes better extract of channel and spatial feature from the features map. The publicly available dataset was used for the experiment comprises 5800 images, among which 3875 were infected with pneumonia, while others were normal images, i.e., 1341. The researchers have used the Convolutional Block Attention Module (CBAM) with the ResNet models which has resulted in the overall effectiveness of disease detection. The epoch of 30 was used in the experiment. While using the attention mechanism, ResNet50, resnet101 and resnet152 have produced accuracies of 90%, 92%, and 94%, respectively, while those taken by each epoch are 22 s, 21 s and 28 s. The result demonstrates that, although the performance of the resnet152 with the attention mechanism performs well in terms of accuracy, time consumption on the other hand has been increased, while ResNet-50 has slightly less accuracy in contrast, the time consumption on each epoch is less. This experiment gives us an understanding of using a model that is capable of producing effective classification on detection of the disease and the least possible time of each epoch. While considering the larger dataset, as in case of real-time, the time element plays an essential role as a higher training time can adversely impact the overall architecture.

An experiment was performed on the classification of malaria cells by using the deep learning methods [18]. The authors have conducted the experiment by using the CNN model of Alexnet, ResNet-50, DenseNet201, vgg19, GoggleNet and Inception 3. In the experimental setup, the authors have used filtration techniques that include medium filter and gauss filter. The highest accuracy of 97.83% was achieved on the DenseNet201 gauss filter settings. Although the use of gauss filter has increased the overall accuracy, gaussian filtering can blur the images which ultimately results in the loss of essential image details including edges, which causes ineffective classification. It is also one of the processor-intensive techniques that would be effectively suitable for deploying in the federated learning environment.

Another interesting experiment was performed that has demonstrated the performance of aggregated deep learning models that includes ResNet-34, VGG-19, DenseNet-121, and DenseNet-161 [19]. The experiment was conducted on the detection of knee osteoarthritis. The dataset used in the experiment was available publicly on Kaggle, and constituted 9786 images in total, which were classed into four different grades. The experiment was conducted individually on the models as well as the with the ensemble approach. The proposed ensemble approach has produced an accuracy of 98%. The authors have used the fine-tuned model to ensemble and evaluate the results. Although, authors have used the effective approach of ensemble for the detection of knee osteoarthritis. The limitation involves the model overfitting issues where different models have their own capabilities to input and process the data. It is important to ensure that model performance is maintained by feeding with the different quantity and variety of the dataset as in the case of a real-time stream of data.

The research [20] was carried out using block chain technology in a federated learning environment. In the proposed study, a novel approach of weight modification was used to train local models from different data sources. The concept of federated learning falls under single point failure; therefore, to cope up with this challenge researchers have used the blockchain in collaboration, which involves the training of the dataset from the different sources based on nodes and ledgers. As utilising the blockchain is immutable, the history of all events is preserved. In the multi-disease classification, the researchers have achieved an accuracy of 88.10%. The experiment has demonstrated the use of federated learning in conjunction with blockchain technology for medical image classification; however, the use of blockchain ledger in the federated learning architecture slows the process of training the model and aggregating at the central server. Therefore, in real time, to effectively use the federated learning architecture, the collaboration of blockchain technology possesses limitations. Our proposed study is based on utilising the solely federated learning architecture and adjusting hyperparameters as well as adding an optimiser to effective medical image classification.

According to the research, there are quite limited data available to study the implications of federated learning [21]. Several experimental studies have been performed during the COVID-19 pandemic while using the framework of horizontal federated learning [22,23,24]. The authors have highlighted the necessity of using the decentralized framework. It was analyzed that due to new pandemic breakout, data are not sufficient to train the local medical institutes without collaboration. The technique of FL has provided the benefit of using data from different hospitals where the uneven concentration of COVID-19 X-rays can be equally used for all participants. Other research has highlighted the importance of using federated learning in cancer detection [25]. The researchers have significantly contributed towards using the decentralized framework effectively, while ensuring data privacy. The study clearly highlights the importance of federated learning in distinct diseases. Our research is inspired by the significance of using the FL framework for medical disease detection.

An experiment was conducted by using the federated transfer learning framework that involves the training of the model among distinct clients that share similar data distribution [26]. Researchers have used this concept to aggregate the class specialty of one client and transferred it to the other clients by mutual collaboration. The proposed framework constitutes clustering mechanisms for higher model efficiency. Another research was conducted by the researchers using the transfer federated learning framework for credit scoring [4]. The experiment was conducted on five distinct datasets and it has been demonstrated an effective approach for credit scoring. Although transfer federated learning has many advantages over the several domains including credit card fraud detection, the process can be biased as it can cause data distribution mismatch and complexity overheads.

We have considered the use of DenseNet and ResNet models on the federated learning architecture as these are the state-of-the-art algorithms used in the image classification as observed in the above literature analysis. The architecture of these models allows the learning of complex image patterns, as in the case of medical images. The skipping connections among the layers in ResNet and interconnected block of DenseNet make these models ideal for learning the data patterns that are not identical among the different clients as in the case of federated learning. The capabilities of transfer learning, as well as performance efficiency in terms of scaling, make ResNet and DenseNet models an effective choice for medical image classification on the real time data. Table 1 shows the brief overview of different literature analyses with respect to our contribution to the knowledge.

## 3. Methodology

### 3.1. Research Model Design

The conventional system adopted by medical institutions and hospitals abides by data protection and privacy laws. This is why data are not shared with other institutions or any third party due to GDPR. Therefore, in our FL framework, data privacy is ensured while model training is performed in multiple medical institutes and hospitals without sharing data. In medical image detection, this approach is effective as it helps train the ML model from various medical centers. In this way, a significant amount of heterogeneous real-time data is used, which ultimately helps the ML model for effective training. The framework of using FL with the CNN-based pre-trained models is selected based on previous work on medical image detection. As mentioned in the literature review, the performance of the ML model is enhanced if the model is trained with a larger amount of data. In other words, fewer data produces ineffective performance in ML model training and vice versa. Therefore, a collaborative way of using the dataset’s multiple sources is required, which can be possible with mutual collaboration. In this way, the model learns the patterns of the image data and helps to form an effective model for detecting diseases from pool of datasets. Figure 1 illustrates our research model design as follows:

As elaborated in the Figure 1, our research workflow follows the data collection. The collected dataset was pre-processed by resizing and removing any irregularities. Once the data are pre-processed, they are split into training, validation, and testing with a ratio of 70%, 20%, and 10%, respectively. The training dataset is used for applying the machine learning model after processing which is sent across the participants for model training in the aforementioned federated learning framework. Once the model is locally trained on the local devices, the individual trained model is shared with the FL server (central server), forming an updated model after aggregation. Secure aggregation is in place in this model design to ensure the security of the trained model. The performance of the model is evaluated with the validation and testing data set. Training the ML model on the individual devices and the FL server is an iterative process to obtain more updates and correspondence from the dataset on the client side. As our research framework is based on FL, let us explore the fundamental concepts behind FL. In the federated learning architecture, the merging algorithm used for the model updates from the distinct client is known as the federated averaging (FedAvg) [27].

We have used FL architecture in our research to demonstrate the privacy-preserving approach that allows for mutual collaboration between hospitals and medical institutions. A standard approach is required to be agreed by participating parties and includes the model framework, loss, and activation functions. In general, the FL architecture can be illustrated as follows [28]:(1)$$\underset{w\in R^{d}}{min}l(x,y;w)\;\mathrm{where}\;l(x,y;w)\overset{\;\mathrm{def}\;}{=}\frac{1}{n}\sum_{i=1}^{n}l\left(x_{i},y_{i};w\right)$$

To understand the architecture of FL, let us take hospitals ‘C’ comprising the dataset ‘Di = nc’, ‘n’ shows the quantity of data, and the FL architecture in the individual hospital can be illustrated as follows [21]:(2)$$l(x,y;w)=\;\mathrm{where}\;L_{c}\left(x_{c},y_{c};w\right)=\frac{1}{n_{c}}\sum_{i\in D_{i}}l\left(x_{i}^{c},y_{i}^{c};w\right)$$

The FL global server inputs the model that consists of different parameters for medical image detection. During the individual cycle with the global server, random participants are considered, then generate the one-to-one link with the server. The local participant downloads the model from the local server that calculates the average gradient based on the loss ‘fc’ which can be demonstrated based on the weight ‘wt’ that constitutes the learning rate ‘η’. In this way, the local participant keeps updating and at the same time keeps updating the central server. The aggregated model receiving the updates from the participants can be calculated as follows [28]:(3)$$\begin{array}{c} w_{t+1}\leftarrow w_{t}-\eta \nabla l(x,y;w)\\ w_{t+1}\leftarrow w_{t}-\eta \sum_{c=1}^{C}\frac{n_{c}}{n}\nabla L_{c}\left(x_{c},y_{c};w\right) \end{array}$$
(4)$$w_{t+1}\leftarrow w_{t}-\eta \frac{n_{c}}{n}f_{c}$$

For every hospital c, $w_{t+1}^{c}\leftarrow w_{t}-\eta f_{c}$, then
(5)$$w_{t+1}\leftarrow w_{t}-\sum_{c=1}^{C}\frac{n_{c}}{n}w_{t+1}^{c}$$

The various sizes of the dataset at each round help to improve the model learning capability in terms of detection $w_{t+1}^{c}$, which can be quite useful for the skewed dataset and can be illustrated in the following equation [21]:(6)$$w_{t+1}\leftarrow w_{t}-\sum_{c=1}^{C}\frac{n_{c}}{n}\alpha_{t+1}^{c}w_{t+1}^{c}$$

The concept of increasing the weight parameter can result in better performance. The averaging of the weight from the distinct client tends to improve the learning capabilities of the model for image detection.

### 3.2. Configuration of Models in Federated Learning

In the configuration setting of our models in the federated learning framework, we have used ‘return_df()’ function that is capable of processing the ‘.png’ images file directories based on distinct datasets and spread equally across the participating clients, i.e., client 4, 6 and 8. The individual file ‘.png’ keeps track of the file location. The return function is capable of performing an equal distribution of available data labels to the clients. For example, if we have five clients and 100 images, then the return function will ensure the distribution of 20 images to individual clients.

In our setting, we have also used custom data loader, which is efficient as it avoids loading the full dataset in RAM and instead loads on demand basis. This is an effective approach, especially on larger datasets. While using the data loading function, i.e., ‘return_dataloader()’, the data loader object is outputted in Figure 2:

8.TrainLoader: It is the type of data loader that determines the training section of the dataset. It is quite useful in iterating the transformed images, as well as in iterating the labels over the batches. Variability is also ensured, as it is involved in the reshuffling of the training data prior to every epoch.9.valLoader: It shows the validation section of the dataset. Data iteration takes place over batches on validation images, as well as labels. It is not involved in the data reshuffling; therefore, the order stays intact throughout the epochs.10.testLoader: It is the type of data loader that demonstrates the test part of the dataset. The data iteration takes place over the batched over the test images as well as labels. Unlike TrainLoader, it is not involved in data reshuffling.

Data loaders are dependant on the percentage allocated for the training, validation, and the testing data. In general, data loaders are quite efficient in memory consumption and can be effectively used in the training validation and testing stages of ML models. It eventually helps to make the data ready for PyTorch-based models.

We have also used the ‘train_model()’ function, which is involved in the following stages:11.Initialise the variable and list: The initial step involved in the tracking of the model state, validation loss, and the accuracy for the individual epoch.12.Epoch loops: The train model function also loops over a certain number of epochs.13.Phase loops: The function also performs the loops over the training as well as the validation stage of individual epochs.14.Set Models Mode: The function also keeps the mode to ‘train’ if the model is in training mode. In this way, certain features including dropout as well as batch normalisation are activated. Similarly, during the validation stage, these parameters are disabled.15.Batch loops: The function also loops with the data batches.16.Forward pass: The function transfers the input label to the corresponding device via model and performs loss calculation.17.Backward pass and optimisation: When using the train model function, if the training is zero, then the gradient of the model is calculated by using the optimisers also known as backward pass.18.Statistics calculation: In the train_model function, the prediction is calculated, and the model run loss as well as accuracy is updated.19.Epoch Statistic Calculation: During the training phase, by the end of every epoch, the loss and accuracy are calculated by the end of every epoch. While in the vase of validation stage, the precision, accuracy, F1-score is calculated while showing the confusion matrix.

Several numbers of iterations are performed by using the while loop. The ‘while loop’ in general is involved in higher number of continuous iterations, however, in our case, it runs at one time based on the rounds:In our case, we have used clients 4, 6 and 8 to train the model individually on the clients while using the ‘train_model’ function, and in this way, model accuracy and loss are given out.The weights from the individual clients are kept in ‘w_local’ lists, similarly, the accuracy and loss of the model are kept on their respective lists.In the federated learning framework, the ‘fed_avg’ function is used to take the average weight of all models to form the global model.The mean loss and accuracy are also calculated on the participants.The mean average of all the weights is referred back towards the model on individual devices. It allows every client to receive the similar model update.The weight average of the models is stored as the ‘fed_model_client’.The output is displayed with the round phase, loss, as well as the accuracy.Alterations in ‘validation_loss’ as well as accuracy from the last round are displayed too.

In this configuration setup of federated learning, we have demonstrated that ML models are locally trained on participating clients, the weight of individual clients is aggregated to form a global model which is shared acres the clients. In this approach, data stay local to client, and only the weight average model is shared to central server. In these data, data privacy is retained.

20.Return best model: Once all corresponding epoch rounds are accomplished, the one with the least validation loss loads up the model.

At every epoch, initially the model is trained with the training dataset, and once the training is done, the model performance is observed with the validation data. The confusion matrix including the accuracy and loss function is demonstrated at each stage to understand the performance of the model. Once all the epochs are completed, the function determines the epoch with the least validation loss, which ultimately selects the corresponding model state. It is a quite useful method that helps us to understand the state of the model in terms of its best performance.

The overall methodology illustrates the different steps from data gathering to exploratory data analysis. Once the pre-processing is completed, data are training over the distinct number of clients in a federated learning environment. Once the data are training, the trained model is aggregated together in central server which also reflects our contribution where the data is not shared to central server, only the trained model is sent across keeping the data privacy. The next section is followed up with data gathering.

### 3.3. Data Gathering

The mobile application was designed by Lister Hill National Center for Biomedical Communication, known as “malaria screener”, and was designed specifically for individuals who work with microscopes but don’t have massive resources. The mobile application was linked to the microscope, where the camera was used to capture pictures. The malaria pictures were marked by professionals from the Mahidol Oxford Tropical Research Institute, which is based in Bangkok, Thailand. The data set comprises 27,560 images which contain those both infected and unaffected [29]. The data set was gathered from the different locations based on the types of malaria as follows:21.Malaria falciparum blood samples were taken from 150 patients at the Chittaging Medical College Hospital, Bangladesh.22.The vivax malaria samples were taken from the same location as above from 150 patients, and also 50 from healthy individuals.23.Malaria samples from falciparum were also taken from a similar location in Bangladesh: 148 patients and 45 healthy individuals.24.Vibrax samples, which is another form of malaria, were also taken from Bangkok, Thailand, from 171 patients.25.In addition, blood cell samples of falciparum malaria were collected from Bangladesh, from 150 patients and others from 50 healthy individuals.

We have used the malaria dataset for several reasons, some of the main reasons are as follows:26.Real-time data, non-synthetic: In the experiments, synthetic data can be quite useful for training the ML models, however, the lab-based model has a limitation to real-time complexity and variance of the disease. The data collected from the real-time provide stronger robust and efficient data for training the models. As in our experiments, we are targeting the real-time data; therefore, the given dataset is collected from the patients in real time, which helps model generalisation.27.Geographical relevancy: Another significant impact of using this dataset involves its correspondence with the geographic relevance where malaria is the serious health issue, i.e., in Bangladesh and Thailand. The geographic location of the data helps to make the effective model prediction based on the provided data. The model customisation facility based on geographic location can help achieve higher performance. Similarly, it assists in nonspecific regions for malaria detection as well.28.Reliable dataset: Another important aspect of using these data is the reliability, as the data have been collected from the endorsed hospitals that maintain the standards. Therefore, higher reliability is essential for better model performance.29.Diverse images: The diverse malaria image collection helps to understand the variations of the causing parasites, which ultimately helps to train the ML model effectively.30.Significant global impact: Malaria is one of the serious diseases that affect millions of individuals each year, with major health consequences. Therefore, the reflection of malaria disease constitutes the main global consideration.

The selection of the malaria dataset fulfils our goal of diagnosing the disease that has a significant impact globally. The work done in the field will help to improve public health and avoid the maximum serious consequences of the disease. 

### 3.4. Exploratory Data Analysis (EDA)

To perform the EDA, we have randomly selected the training dataset to visualise the data diversity. We have taken 15 images individually from the training dataset and the validation dataset with their own labels. The visualisation of the training dataset has helped us to understand the data, while observing its performance during the validation stage. We have divided the dataset for training, validation and testing in the ratio of 70%, 20%, and 10%, respectively. The EDA on the malaria dataset is quite important as it helped to ensure the correct labels on the images, and it also aids in understanding the data variation and complexities, so the models can be adjusted accordingly while training.

#### 3.4.1. Number of Available Datasets

The individual dataset that is available on the training and validation dataset is 13,780. The count of dataset, based on the infected and unaffected, can be visualized in Figure 3:

The total count of 27,560 images is available when combining the training and validation datasets. The large number of images helps our models to perform an effective training from the variety of images. The balance of the dataset helps to achieve better model training that is essential for correct disease prediction.

#### 3.4.2. How Parasitised and Nonparasitised Cells Look

When observing the malaria dataset, the main element that makes the difference between the two classes is the ‘dots’ that are visible in the parasitised (infected) class. The dots are actually the malaria parasite. The parasite that causes malaria is known as the plasmodium, which directly targets red blood cells of the body. Under the microscope, the infected part of the cell is seen as small dots.

In our dataset, we can see the ‘ting dots’ that are represented in the pink color as the parasite in the cells, which is one of the visual sides of detecting malaria. The tiny dots also make a good difference between the healthy cells, in contrast to those affected by the malaria parasite. Healthy cells lack these dots that make up non-malaria cells. The difference between healthy cells and malaria cells can be identified in Figure 4 as follows

### 3.5. Data Preparation

Data preparation is an initial step to prepare the data for machine learning modelling. Therefore, the images must be changed in the form and shape that the computer could understand. The following are the steps involved in our data preparation.

The initial stage involves reading the images from the directory.Decoding of the image content that involves converting into grid form as RGB.Conversion of images into float point tensor.Rescaling the tensors into the form that allows the scale range from 0 and 255 to be 0 s and 1 s as the CNN models takes in the smaller inputs.

We have performed the above steps using the TensorFlow tool known as the ImageDataGenerator. It helps to form the images in the required form for CNN models. The sampling of images also takes place with a similar tool. Afterwards, we used the technique known as the flow-from-directory, which helps to input the images, adjust the value, and rescale the size of the images. The significance of data pre-processing involves the conversion of images into the shape and size so that it can be compatible to effectively perform on the CNN-based pre-trained models of DenseNet and ResNet-50.

### 3.6. Visualize the Training Images

We have taken 15 random images from the training dataset, as well as from the validation dataset, which can be visualised in Figure 5 and Figure 6 with respect to their labels:

## 4. Malaria Experiments and Results

### 4.1. FL_DenseNet and FL_ResNet-50 (4 Clients)

The Figure 7 below illustrates the confusion matrix for DenseNet and ResNet-50 with four clients:

In terms of performance of model towards the unseen data, let us consider the test data performance analysis with critical discussion:31.Accuracy: (TP + TN)/(TP + FP + FN + TN)

With the overall correct malaria predictions out of all predictions, the accuracy can be calculated as follows:DenseNet: (1276 + 1353)/(1276 + 1353 + 39 + 84) = 0.9463 or 94.63%ResNet-50: (1254 + 1366)/(1254 + 1366 + 26 + 106) = 0.9486 or 94.86%

ResNet-50 has slightly higher accuracy than DenseNet.

32.Precision: TP/(TP + FP)

It shows the accuracy of the positive prediction of malaria and can be observed in the following equation:DenseNet: 1276/(1276 + 39) = 0.9703 or 97.03%ResNet-50: 1254/(1254 + 26) = 0.9796 or 97.96%

ResNet-50 has slightly higher precision.

33.Recall (sensitivity): TP/(TP + FN)

It involves the fraction of positive predictions which is correctly determined:DenseNet: 1276/(1276 + 84) = 0.9382 or 93.82%ResNet-50: 1254/(1254 + 106) = 0.9220 or 92.20%

DenseNet has slightly higher recall.

34.Specificity: TN/(TN + FP)

It involves the fractions of negative prediction that are correctly determined:DenseNet: 1353/(1353 + 39) = 0.9720 or 97.20%ResNet-50: 1366/(1366 + 26) = 0.9813 or 98.13%

ResNet has higher specificity.

35.F1 Score: 2 × (Precision × Recall)/(Precision + Recall)

The weighted average of precision and recall of both models can be calculated as follows:DenseNet: 2 × (0.9703 × 0.9382)/(0.9703 + 0.9382) = 0.9540 or 95.40%ResNet-50: 2 × (0.9796 × 0.9220)/(0.9796 + 0.9220) = 0.9501 or 95.01%

DenseNet has a slightly higher F1 score.

The evaluation metric of the above results can be illustrated in the Table 2:

It can be observed from the above analysis that DenseNet has higher recall as well as F1-score, however, ResNet-50 stands out with its accuracy, precision, and specificity.

To determine the selection of model based on the above results, it depends on the use case in any given situation. In this case, if we need to decrease false negative values, for example to classify if someone who does not have malaria actually has it, the selection of DenseNet is preferred, as it has a higher recall rate. In the other case, if we need to reduce false positive values, in other words, if the model tells someone they have malaria, however in reality they do not, then ResNet-50 could be a better option due to better precision and specificity. ResNet-50 is also leads slightly in accuracy. The following trend can be observed in Figure 8 and Figure 9 considering the train/validation loss on device 1, 2, 3, and 4 (malaria dataset—4 clients) DenseNet and ResNet:

It is worthwhile to understand the importance of model selection based on the metric; however, the selection mostly relies on the use case scenario. In real life, it is good to have a model with few false positive values (including some further testing), instead of missing true positive results that can prevent treatment. Therefore, in this case, the model with higher recall rate should be prioritised.

### 4.2. FL_DenseNet and FL_ResNet-50 (6 Clients)

The following confusion matrix is achieved by using 6 clients in the malaria dataset:36.Accuracy:
DenseNet: (688 + 1272)/(688 + 0 + 168 + 1272) = 0.9213 (92.13%)ResNet-50: (666 + 1269)/(666 + 3 + 703 + 1269) = 0.7250 (72.50%)
37.Precision:
DenseNet: 688/(688 + 0) = 1 (100%)ResNet-50: 666/(666 + 3) = 0.9955 (99.55%)
38.Recall:
DenseNet: 688/(688 + 168) = 0.8037 (80.37%)ResNet-50: 666/(666 + 703) = 0.4864 (48.64%)
39.F1-score:
DenseNet: 2 × (1 × 0.8037)/(1 + 0.8037) = 0.8911 (89.11%)ResNet-50: 2 × (0.9955 × 0.4864)/(0.9955 + 0.4864) = 0.6530 (65.30%)
40.Specificity:
DenseNet: 1272/(1272 + 0) = 1 (100%)ResNet-50: 1269/(1269 + 3) = 0.9976 (99.76%)

The evaluation metric of the above results can be illustrated in Table 3:

The above results show the better performance of DenseNet in contrast to ResNet-50 while comparing accuracy, recall, f1-score and specificity. ResNet-50 also constitutes less precision, while the difference is quite minor. The overall comparison shows that there are less false negative and non-false positive in contrast. The detailed analysis shows that the DenseNet model stands out in terms of malaria classification; however, it is also important to understand the certain requirement of the use case. In the case where higher false positives are required, the use of DenseNet could be a good choice, as it constitutes higher precision. Overall, DenseNet performed well in the malaria case study which can be observed in Figure 10 as follows:

### 4.3. FL_DenseNet and FL_ResNet-50 (8 Clients)

We have evaluated the models’ performance based on the following matrix:41.Accuracy:
DenseNet: (1308 + 712)/(1308 + 552 + 68 + 712) = 0.7504 (75.04%)ResNet-50: (666 + 1269)/(666 + 3 + 703 + 1269) = 0.7250 (72.50%)

The accuracy result shows that the DenseNet performs well in accuracy as compared to DenseNet demonstrating the fewer errors. On the other case, the only accuracy is not enough to justify the model performance in the case where the rate of false positive is massively different from that of false negative.

42.Precision:

DenseNet: 1308/(1308 + 552) = 0.7033 (70.33%)ResNet-50: 666/(666 + 3) = 0.9955 (99.55%)

In the consideration where the cost of false positive is higher, which means the prediction of malaria disease while its reality is not, then ResNet-50 is the best choice.

43.Recall (sensitivity):

DenseNet: 1308/(1308 + 68) = 0.9506 (95.06%)ResNet-50: 666/(666 + 703) = 0.4864 (48.64%)

In the case where it is necessary to determine the actual malaria case while risking the false alarm, the DenseNet model is effective due to the higher recall value. The use of a recall matrix could be ideal in the health care sector where disease identification is crucial.

44.F1-Score:

DenseNet: 2 × (0.7033 × 0.9506)/(0.7033 + 0.9506) = 0.8087 (80.87%)ResNet-50: 2 × (0.9955 × 0.4864)/(0.9955 + 0.4864) = 0.6530 (65.30%)

Determines the relationship between precision and recall values. In other words, f1-score plays an essential role where the false positive as well as the false negative are of equal importance. In our case, the DenseNet model is the best selection for f1-score.

45.Specificity (True Negative Rate):

DenseNet: 712/(712 + 552) = 0.5635 (56.35%)ResNet-50: 1269/(1269 + 3) = 0.9976 (99.76%)

The evaluation metric of the above results can be illustrated in the Table 4:

Figure 11 is the graphical illustration with eight clients on DenseNet:

Specificity is quite useful in the cases where it is required to determine the negative cases that do not constitute malaria. In our case, ResNet-50 performs a higher specificity score, which is useful in the situation where it requires one to avoid unnecessary treatment. Figure 12 shows the confusion metrix based on 8 clients.

### 4.4. Significance Test

The *t*-test suggests whether the difference between the methods really makes the difference based on the critical differences, or whether it is just a coincidence [30]. Therefore, we have used our CNN models along with FL to conduct pairwise tests. The *p*-value reflects whether the difference between the models is larger, in case if the values go lower than 5%, it will result in rejection. The *p*-test conducted on the CNN models individually on the FL framework reflects the performance based on each model. In other words, the *t*-test shows the mean difference of the chosen parameter between the two models. The positive t-values indicate whether the first group have any differences with the second group and negative values reflect that the second group has major differences with the first one. Similarly, the *p*-value, as in our case, is standard scale of 0.05, so if the results are below this value it indicates the significant difference between the models. The following Table 5 shows the t-stats and *p*-values of our results:

## 5. Discussion

The performance of the model’s comparison using the four clients shows that the DenseNet constitutes higher recall as well as F1-score while on the other hand, ResNet-50 shows higher accuracy, precision and specificity. If we observe the performance of the models under six clients, it can be seen that the ResNet-50 shows less precision, however the difference is minor. It can be observed that the performance of the DenseNet stands out in classification. Depending on the type of use in cases, both models play an essential role in malaria image detection as well as classification. While performing the experiments under eight clients, it was observed that the DenseNet performed quite well in classification, in contrast to ResNet-50.

Based on the models’ performance, the following situations can be taken into account:In the case where it is necessary to determine actual malaria disease as much as possible, DenseNet is preferred due to higher recall.In the case where it is required to determine actual malaria disease, and in reality, it is malaria, then ResNet-50 is preferred due to its higher precision.In the case where it is required to determine the balancing among the false positive as well as false negative, then DenseNet performs well due to higher recall.In the case where it is required to correctly determine the negative cases, that is, no-malaria, ResNet-50 performs well due to its effective specificity results.

In overall experiments, it was observed that the classification accuracy reduced as the number of clients are increased. One of the reasons for this is the limited dataset availability, as the dataset is divided across various participants. However, in a stream of real-time dataset, where the high availability of data would eventually divide into distinct number of clients, it will enhance the classification performance.

## 6. Summary

While observing the above results, it can be seen that the DenseNet model performs better in terms of accuracy, recall, and f1-score, while on the other hand, ResNet-50 did well in precision and specificity. We have experimented both models in different settings and our preliminary results showed that the DenseNet model performed better in accuracy (75%), in contrast to ResNet-50 (72%), when considering 8 clients, while the trend is observed common in four clients with the similar accuracy of 94% and use in six clients showed that the DenseNet model performed quite well with the accuracy of 92%, while ResNet-50 achieved only 72%.

In general, DenseNet might be the best choice if the accuracy of predicting the malaria cases is crucial, however, the ResNet-50 model avoids false alarms and also helps in the correct detection of non-malaria cases. The use of federated learning architecture among the distinct clients for ensuring the data privacy and following the GDPR is the contribution of this research work.

## 7. Limitation and Constraints

Although the use of ResNet and DenseNet have shown good results in image classification, there are limitations as well. As the medical images are complex in nature, once the CNN-based pre-trained models are used, it degrades the image size that can cause the loss of essential features and ultimately results in inefficient accuracy. CNN models can also be inefficient on the larger-scale datasets, especially while training the model. If the data are wrongly labelled, which could be caused by human error, this can lead to inefficient model training, where results could produce higher numbers of false positives and false negatives. Legal regulations on deploying this architecture in real-time could be challenging, as it requires standard operating procedures. The reliance on image annotation from the medical experts could make this process slower.

## Figures and Tables

**Figure 1 bioengineering-11-00340-f001:**
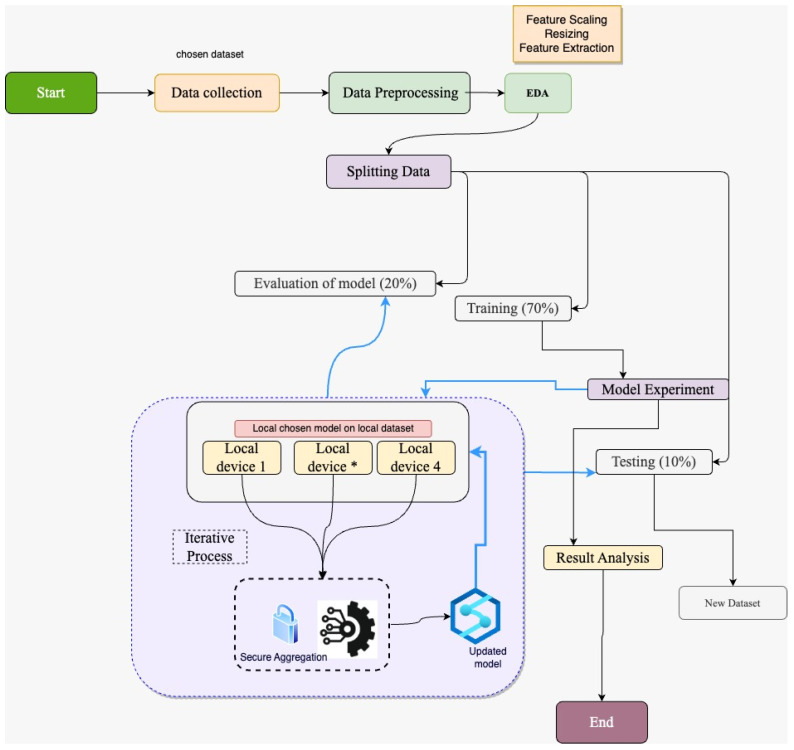
Proposed framework showing the hybrid architecture constitute of CNN based pre-trained model in federated learning. “Local device*”, where “*” shows the device 2 and device 3.

**Figure 2 bioengineering-11-00340-f002:**
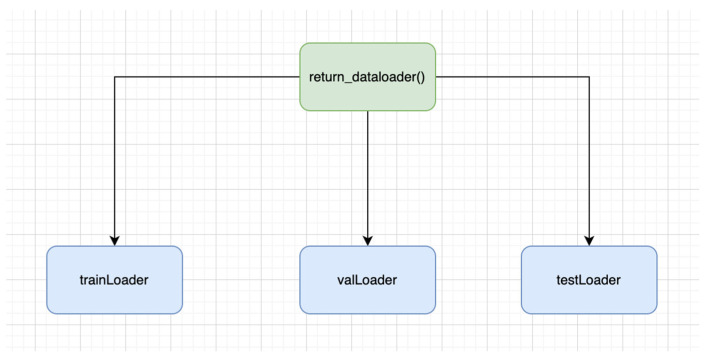
Data loader function.

**Figure 3 bioengineering-11-00340-f003:**
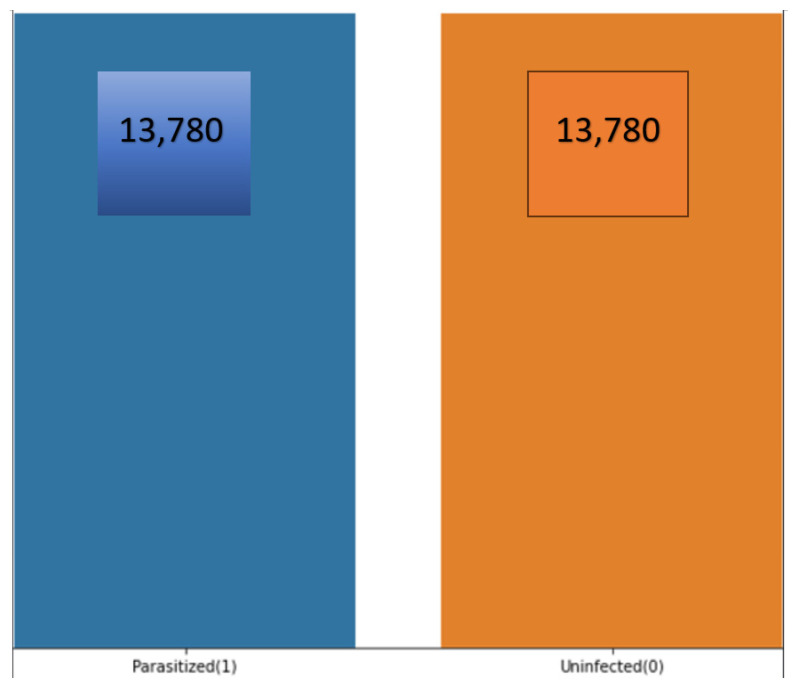
Malaria dataset distribution.

**Figure 4 bioengineering-11-00340-f004:**
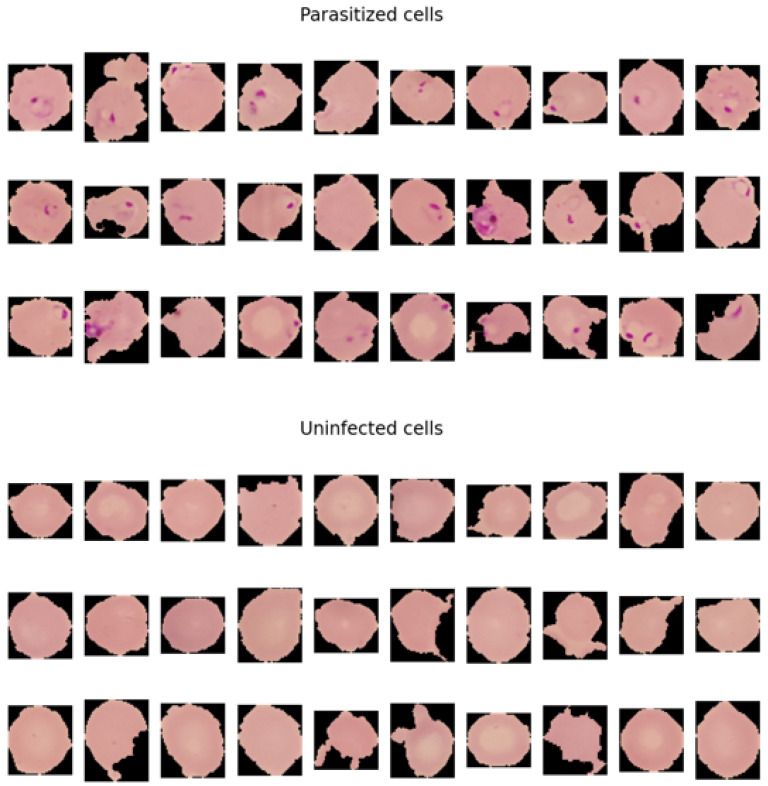
Distinguishing between the cells among the two classes of help models, i.e., resent50 and DenseNet, to learn the patterns from the cells. Therefore, the ML models classify cells with the presence of dots as parasitised and those without dots as uninfected.

**Figure 5 bioengineering-11-00340-f005:**
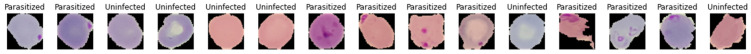
Random images of the sample training images.

**Figure 6 bioengineering-11-00340-f006:**
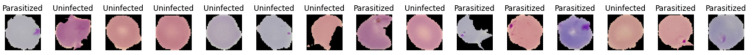
Images from the sample validation images.

**Figure 7 bioengineering-11-00340-f007:**
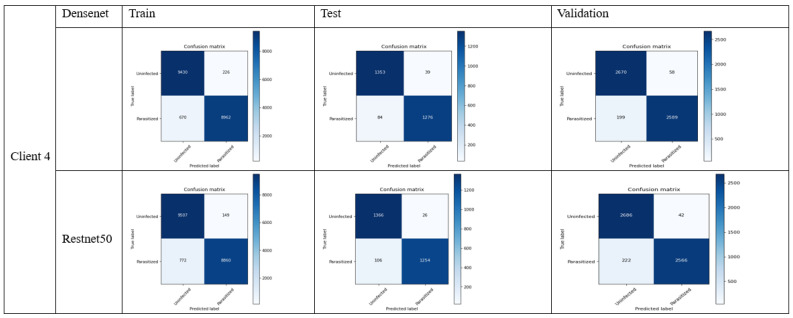
The confusion matrix for DenseNet and ResNet-50 with 4 clients.

**Figure 8 bioengineering-11-00340-f008:**
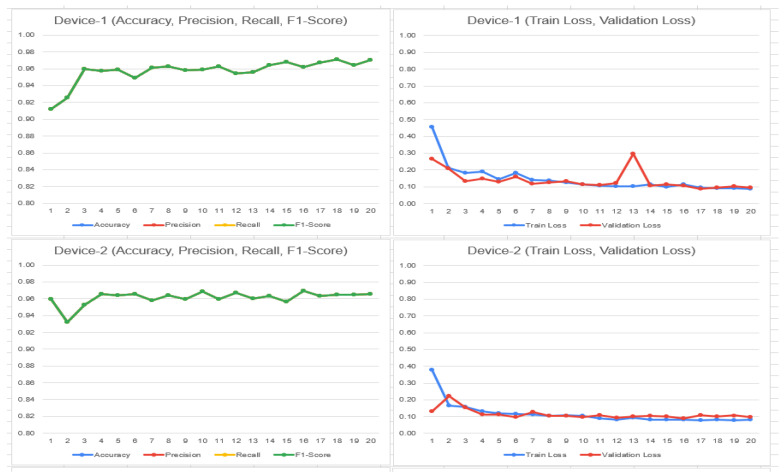
Train and validation loss on ResNet-50 on 4 clients.

**Figure 9 bioengineering-11-00340-f009:**
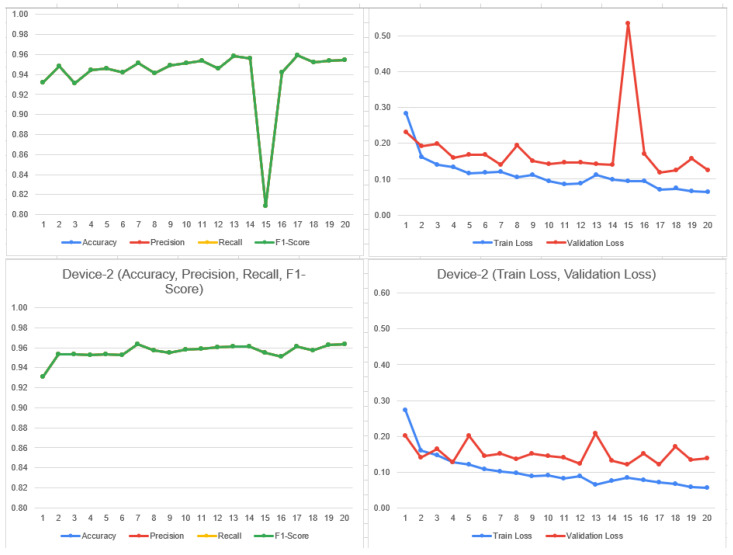
Train and validation loss on DenseNet on 4 clients.

**Figure 10 bioengineering-11-00340-f010:**
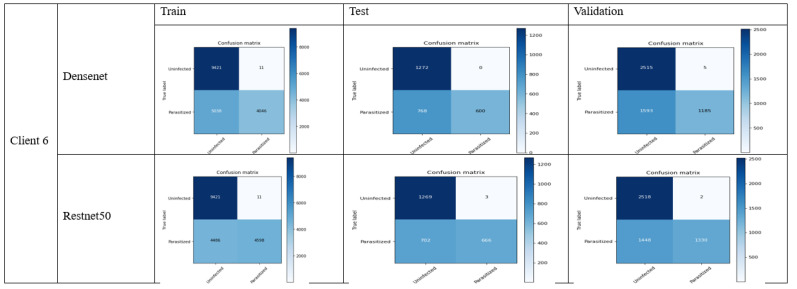
Confusion matrix of FL_DenseNet and FL_ResNet-50 (6 Clients).

**Figure 11 bioengineering-11-00340-f011:**
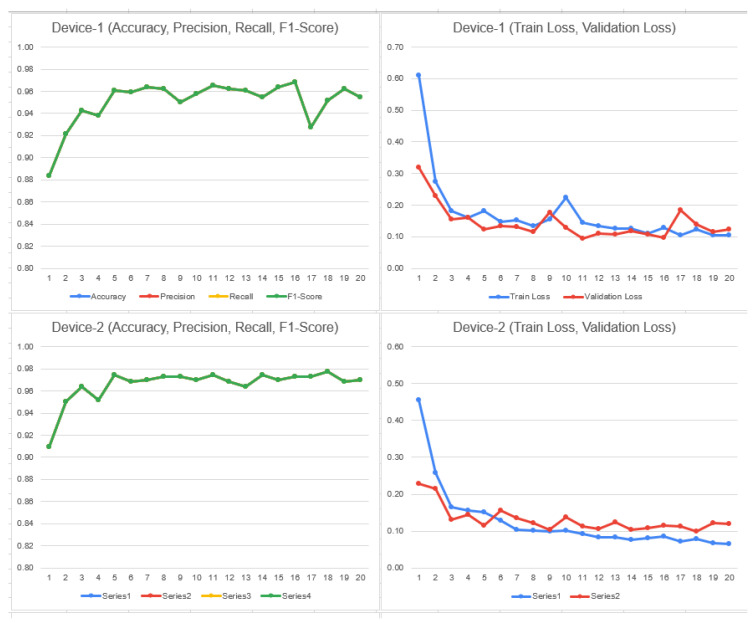
Train and validation loss on DenseNet on 8 clients.

**Figure 12 bioengineering-11-00340-f012:**
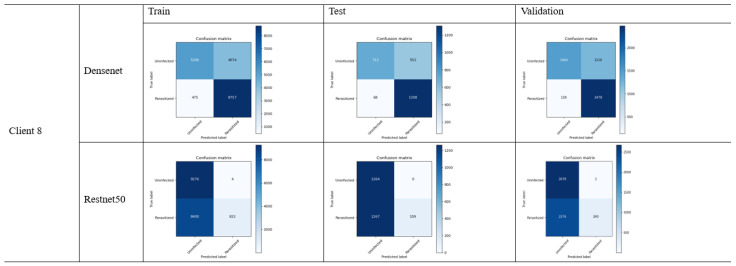
Confusion matrix of FL_DenseNet and FL_ResNet-50 (8 Clients).

**Table 1 bioengineering-11-00340-t001:** Literature review analysis in contrast to the proposed method.

References	Disease	Algorithm	Accuracy	Dataset Size	Pros	Cons
[11]	Diabetic Retinopathy	ResNet-50	98%	3762 images	ResNet-50 is effective when the image pre-processing is enhanced as the model performance can vary depending on the input images.	The performance of this algorithm is not further explored in larger datasets.
[12]	COVID-19	ResNet-50 and VGG16	ResNet50 was 88% VGG16 was 85%	10 k	Effective classification considering both algorithms.	Data is imbalanced. In case of ResNet-50 precision is 100% while for VGG16 its 85%
[13]	COVID-19	ResNet-50	88% to 94%	5 K CT scan	Authors have clearly distinguish the use of different hyperparameter tuning to effectively increase the model accuracy for the disease prediction	Limited dataset size. Hyper tuning could be enhanced with use of some optimizer.
[14]	Tuberculosis (TB)	DenseNet	98.8%	5 K	This research has highlighted the use of different epochs to understand the best possible outcome of the experiment for medical image detection.	Dataset is highly imbalance between the normal and infected images.
[16]	SARS-CoV-2	customised CNN model	92%	2481 CT scan	The customised CNN approach has produced in the classification of the SARS-CoV-2 virus	Limited dataset size. Higher number can adversely impact on the performance of the model.
[17]	Pneumonia	ResNet-50	90%	5800 images	The performance of the ResNet-50 with the attention mechanism performs well in terms of accuracy	Higher time consumption in training dataset
[18]	Malaria	CNN models	97.83% for DenseNet-201	6730 images	The use of gauss filter has increased the overall accuracy	Processor intensive. Gaussian filtering can blur the images which ultimately results in the loss of essential image details including edges which causes ineffective classification.
[19]	Knee Osteoarthritis	CNN models	98%	9786 images	Effective approach of ensemble for the detection of knee osteoarthritis.	Model overfitting issues where the different models have their own capabilities to input and process the data.
Our contribution	Malaria (4 clients)	ResNet-50 DenseNet	ResNet-50 = 94.86% DenseNet = 94.63%	27 k images	Privacy preserving approach of using dataset from different clients while following GDPR with enhanced accuracy in detection of malaria disease.	Legal regulation of deploying this architecture in the real-time could be challenging as it requires standard operating procedures. The reliance on image annotation from the medical experts can make this process slower.
Our contribution	Malaria (6 clients)	ResNet-50 DenseNet	ResNet-50 = 72.50% DenseNet = 92.13%	27 k images	Privacy preserving approach of using dataset from different clients while following GDPR with enhanced accuracy in detection of malaria disease.	Legal regulation of deploying this architecture in the real-time could be challenging as it requires standard operating procedures. The reliance on image annotation from the medical experts can make this process slower.
Our contribution	Malaria (8 clients)	ResNet-50 DenseNet	ResNet-50 = 72.50% DenseNet = 75.04%	27 k images	Privacy preserving approach of using dataset from different clients while following GDPR with enhanced accuracy in detection of malaria disease.	Legal regulation of deploying this architecture in the real-time could be challenging as it requires standard operating procedures. The reliance on image annotation from the medical experts can make this process slower.

**Table 2 bioengineering-11-00340-t002:** Evaluation metrics based on 4 clients.

	Accuracy	Precision	Recall	F1 Score	Specificity
ResNet-50	96.63%	97.96%	92.20%	95.01%	98.13%
DenseNet	94.86%	97.03%	93.82%	95.40%	97.20%

**Table 3 bioengineering-11-00340-t003:** Evaluation metrics based on 6 clients.

	Accuracy	Precision	Recall	F1 Score	Specificity
**ResNet-50**	72.50%	99.55%	48.64%	65.30%	100.00%
**DenseNet**	92.13%	100.00%	80.37%	89.11%	97.20%

**Table 4 bioengineering-11-00340-t004:** Evaluation metrics based on 8 clients.

	Accuracy	Precision	Recall	F1 Score	Specificity
**ResNet-50**	72.50%	99.55%	48.64%	65.30%	99.76%
**DenseNet**	75.04%	70.33%	95.06%	80.87%	56.35%

**Table 5 bioengineering-11-00340-t005:** T-stats and *p*-values of different models on malaria dataset.

Metrics	Model Name	T-Stats	*p*-Value
Accuracy	FL_DENSENET And FL RESNET-50	11.45726	0
Precision	FL_DENSENET And FL RESNET-50	−0.09118	0.92746
Re-call	FL_DENSENET And FL RESNET-50	0.09637	0.92335
F1-Score	FL_DENSENET And FL RESNET-50	0.25665	0.79778

## Data Availability

The data used for the manuscript has been gathered from the publicly available source i.e., Kagle, link: https://www.kaggle.com/datasets/iarunava/cell-images-for-detecting-malaria.

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
