# Peer review of "A Privacy-Preserving Approach to Effectively Utilize Distributed Data for Malaria Image Detection"

_bioengineering, 2024, doi:10.3390/bioengineering11040340_

Round 1

Reviewer 1 Report (Previous Reviewer 2)

Comments and Suggestions for Authors

I re-examined your work titled "A Privacy-Preserving Approach to Effectively Leverage Distributed Data for Malaria Image Detection" in detail. It appears that the work has been significantly improved compared to the previous round. It was observed that the Abstract section met the expectations after the revision. It should also be taken into account which algorithm is used in the merging process. Formulas are not presented in the performance measurement metrics calculated after Figure 7, but these formulas are presented after Figure 8. It would be better to present these formulas where they were first used. In the study, it is presented in Figure 1 line 301. However, after Figure 8, it was presented again in a different way as Figure 1. Additionally, figures should be cited in the text. For example, the proposed model is presented in figure 1. This is valid for all figures. Similarly, figure numbers should be reviewed for all figures and cited in the text. The performance evaluation metrics presented in Title 3.2 should be presented in a short table, there is no need to write this long. This table will also make comparison easier. A discussion section has been added to the study. But this section consists of 4 items. You need to rewrite the relevant section in detail. Figure 3 chart showing data numbers is unnecessary. Finally, to compare the success of Federated Learning, the original results of Resnet and Densenet pre-trained models should also be presented. In this way, it will be easier for us to observe the performance of the model you suggest.

Comments on the Quality of English Language

Spelling and grammatical errors should be reviewed.

Author Response

Reviewer 2 Report (New Reviewer)

Comments and Suggestions for Authors

Amer Kareem et al. reported an interesting work about malaria image detection. The topic was to some degree of significance, and could arouse a certain impact in its field. Overall, the submission fell within the scope of Bioengineering. The reviewer suggested a Major Revision before a final acceptance.

Detailed comments:

1)     The Abstract was a bit long. Please consider to shorten it to ~250 words.

2)     For Section 1.2 and 1.3, it would be advisable to express the content in the pattern of “research questions (RQs)”.

3)     The Table should be accompanied with table legends (captions).

4)     The Discussion Section should be substantially expanded to showcase more valuable information, including the clinical translation aspects.

5)     The dataset, if possible, could be uploaded as a Supplementary File to this article.

6)     Please double-check the format of References.

Round 2

Reviewer 1 Report (Previous Reviewer 2)

Comments and Suggestions for Authors

Resnet50 and densenet should be written in the same format in the article. In some places, attention should be paid to the lowercase and uppercase letters. Confusion matrices should be drawn more beautifully.

Comments on the Quality of English Language

Spelling and grammatical errors need to be reviewed.

Reviewer 2 Report (New Reviewer)

Comments and Suggestions for Authors

Thanks for your revision.

This manuscript is a resubmission of an earlier submission. The following is a list of the peer review reports and author responses from that submission.

Round 1

Reviewer 1 Report

Comments and Suggestions for Authors

A Privacy-preserving Approach to Effectively Utilize Distributed Data for Malaria Image Detection

In the abstract, the authors have mentioned the problem but how their working on solving that issue is not clear. Is this work helping to solve the issue? Please correlate it properly. The dataset used?? Methodology novelty. Please discuss these issues.

The reference citation in the introduction is starting with [10]. It should start with [1] and follow the sequence.

In the introduction specify the current statistics of this disease in this study with proper citations. The introduction section should consist of the following: Statistics, Problem Background, State-of-the-art techniques used to counter the problem, Graphical abstract, List of significant contributions, and Paper organization.

Section 2 can be included in the section 1.

The literature review should be supported with a table with a summary of all the state-of-the-art techniques with limitations for each citation.

Section 3 should be about the dataset. Description of the dataset Technical details, Number of samples, and any pre-processing technique applied. What is the significance of the pre-processing techniques???

Training and testing sample details.

Equations are not cited in the text.

Section 4.3 should show some samples of the dataset.

Figure are not properly cited at appropriate places.

Results are not properly justified with loss and accuracy graphs.

No discussion section.

No comparative analysis.

No novelty in methodology.

Reference are very less.

Comments on the Quality of English Language

No comments

Reviewer 2 Report

Comments and Suggestions for Authors

I examined your work titled "A Privacy-preserving Approach to Effectively Utilize Distributed Data for Malaria Image Detection" in detail. Some points in the study should be seriously highlighted. The contributions of federated learning should be highlighted in the study. This part should be highlighted especially in the abstract section. What is the reason for the high accuracy value in federated learning? Here, it is clear that there will be an advantage in speed since the data set is divided into 4 or more clients. It should also be considered which algorithm is used for the merging process. There are many studies on malaria in the literature. Please also review the study titled "Classification of malaria cell images with deep learning architectures" and evaluate the accuracy, sensitivity, etc. Presenting the metrics in a table will make the article more fluid. Additionally, a paragraph regarding the organization of the article should be added. Why are Resnet and Densenet models preferred? Presenting your studies in the literature review and the results of your own study in a table in the discussion section will better reveal the importance of federated learning. Limitations of the study should also be presented in the Discussion section. As a result, it is important to highlight federated learning.

Comments on the Quality of English Language

It is important to review spelling and grammatical errors.

Reviewer 3 Report

Comments and Suggestions for Authors

The article could be of interest. I went through a initial reading of the manuscript but or either the authors sent a wrong draft or submitted a careless version of the manuscript that does not fulfill the quality of this journal. 

Just to mention some issues in a first reading:

"Malaria is one of the life-threatening disease caused by the Anopheles mosquitoes". Malaria is caused by Plasmodium falciparum and not by mosquitoes. Mosquitoes transmit malaria.

"The parasite that causes malaria is known as the plasmodium..."  Plasmodium spp. Plasmodium is the cientific species name of the malaria parasite that causes malaria. Therefore is written with a capital P and italic.

"...we see the ‘ting dots’ as parasite in the cells..." I do not understand this sentence.

"....Vibrax malaria samples..." no idea what this is.

Comments on the Quality of English Language

Needs serious revision.